# Association of Obstructive Sleep Apnea and Atrial Fibrillation in Acute Ischemic Stroke: A Cross-Sectional Study

**DOI:** 10.3390/jpm13030527

**Published:** 2023-03-15

**Authors:** Valerio Brunetti, Elisa Testani, Anna Losurdo, Catello Vollono, Aldobrando Broccolini, Riccardo Di Iorio, Giovanni Frisullo, Fabio Pilato, Paolo Profice, Jessica Marotta, Eleonora Rollo, Irene Scala, Paolo Calabresi, Giacomo Della Marca

**Affiliations:** 1UOC Neurologia, Dipartimento di Neuroscienze, Organi di Senso e Torace, Fondazione Policlinico Universitario A. Gemelli IRCCS, Largo A. Gemelli, 8, 00168 Rome, Italy; 2Department of Neurosciences, Università Cattolica del Sacro Cuore, Largo Francesco Vito, 1, 00168 Rome, Italy; 3Stroke Unit, Azienda Ospedaliera San Camillo, 00152 Rome, Italy; 4Servizio di Neurologia e Neurofisiopatologia, Humanitas San Pio X, 20159 Milan, Italy; 5UOC Neurofisiopatologia, Dipartimento di Neuroscienze, Organi di Senso e Torace, Fondazione Policlinico Universitario A. Gemelli IRCCS, Largo A. Gemelli, 8, 00168 Rome, Italy; 6UOC Neurologia, Università Campus Bio-Medico di Roma, Via Alvaro del Portillo, 200, 00128 Rome, Italy; 7UOC Neurologia and Stroke Unit, Mater Olbia Hospital, 07026 Olbia, Italy

**Keywords:** atrial fibrillation, central sleep apnea, cerebrovascular diseases, obstructive sleep apnea, OSA, sleep, sleep-disordered breathing, stroke, stroke unit

## Abstract

*Background*: There is a growing body of evidence suggesting a link between obstructive sleep apnea (OSA) and atrial fibrillation (AF). The primary objective of this study is to evaluate the association between OSA and AF in acute ischemic stroke. The secondary objective is to describe the clinical features of patients with acute ischemic stroke and concomitant OSA. *Methods*: We enrolled consecutive patients with acute ischemic stroke. All patients underwent full-night cardiorespiratory polygraphy. To determine if there is an association between AF and OSA, we compared the observed frequency of this association with the expected frequency from a random co-occurrence of the two conditions. Subsequently, patients with and without OSA were compared. *Results*: A total of 174 patients were enrolled (mean age 67.3 ± 11.6 years; 95 males). OSA and AF were present in 89 and 55 patients, respectively. The association OSA + AF was observed in 33/174 cases, which was not statistically different compared to the expected co-occurrence of the two conditions. Patients with OSA showed a higher neck circumference and body mass index, a higher prevalence of hypertension and dysphagia, and a higher number of central apneas/hypoapneas. In the multivariate analysis, dysphagia and hypertension were independent predictors of OSA. A positive correlation was observed between OSA severity, BMI, and neck circumference. The number of central apneas/hypoapneas was positively correlated with stroke severity. *Conclusions*: Our data suggest that OSA and AF are highly prevalent but not associated in acute stroke. Our findings support the hypothesis that OSA acts as an independent risk factor for stroke.

## 1. Introduction

Obstructive sleep apnea (OSA) and atrial fibrillation (AF) are both highly prevalent in acute ischemic stroke (AIS). There is a significant overlap between OSA and AF, suggesting a possible pathogenic link between these two conditions [1]. Growing evidence suggests that OSA plays a causal role in the initiation and maintenance of AF [2]. OSA can promote arrhythmogenesis through several mechanisms. First, the hemodynamic changes induced by repetitive forced inspiration against the collapsed airway lead to increased cardiac load, which, in turn, leads to atrial remodeling, promoting arrhythmia [3]. Furthermore, repetitive apneas/hypopneas, and the consequent intermittent hypoxemia, induce a metabolic derangement that causes oxidative stress and, consequently, a chronic inflammation condition that increases the susceptibility to develop AF [4]. Finally, OSA results in autonomic imbalance with increased tone and surges of sympathetic activation, providing a favorable substrate for the development of cardiac arrythmias, in particular AF [5]. AF is the most common cause of cardioembolic stroke, representing approximately 15–30% of all ischemic strokes [6]. Recent data indicate that OSA is associated with increased risk of cardioembolic stroke [7], suggesting a role of OSA in determining AIS through AF. Based on these findings, we hypothesized that these conditions are closely associated in AIS. Few studies have investigated the association between AF and OSA in AIS, with inconsistent results [7,8,9,10].

Therefore, the primary endpoint of the current study was to evaluate the association between OSA and AF in AIS, comparing the observed frequency of this association with the expected frequency. The secondary endpoint was to describe the clinical features of patients with AIS and concomitant OSA.

## 2. Materials and Methods

### 2.1. Patients and Data Source

This is a cross-sectional study with prospective enrollment, conducted at the stroke unit of the Gemelli Hospital in Rome. The inclusion criteria were age ≥18 years and diagnosis of ischemic stroke with NIHSS ≥1 confirmed with neuroimaging (brain MRI or CT). Exclusion criteria were patients with an unstable clinical condition and a pre-existing diagnosis of sleep disorder, including OSA. The following data were collected: age, sex, body mass index (BMI), neck circumference, obstructive apnea–hypopnea index (O-AHI), central apnea–hypopnea index (C-AHI), oxygen desaturation index (ODI), AF (from medical history or newly diagnosed), hypertension, diabetes, dyslipidemia, reperfusion therapy (intravenous thrombolysis or endovascular thrombectomy), wake-up stroke, the Oxfordshire Community Stroke Project (OCSP) classification based on clinical symptoms, the etiology of stroke according the Trial of ORG 10172 in Acute Stroke Treatment (TOAST) classification, National Institutes of Health Stroke Scale (NIHSS) at admission, dysphagia, pneumonia, and death. All patients underwent a polygraphic sleep study within 7 days of AIS onset, in an in-hospital setting. The following parameters were recorded: airflow (measured by nasal cannula), thoracic and abdominal effort, snore, and peripheral oxygen saturation. OSA was diagnosed when the obstructive apnea–hypopnea index (O-AHI) was ≥10 events per hour. The choice of this cut-off was based on the meta-analysis of Johnson and Johnson [11], since most of the studies that have assessed the prevalence of OSA in patients with acute stroke in a hospital setting have used an AHI cut-off of ≥10 events/hour. Respiratory events were scored according to the criteria of the American Academy of Sleep Medicine [12]. Sleep apnea was scored when there was a drop of ≥90% of the peak of airflow signal for more than 10 s. Sleep hypopnea was scored if there was a reduction of ≥30% of the airflow for more than 10 s in association with a drop of hemoglobin saturation of ≥3%. Central apneas and hypopneas were not included in the calculation of O-AHI but were considered separate events and included in the count of the central apnea–hypopnea index (C-AHI).

Continuous multiparametric monitoring (electrocardiogram, hemoglobin saturation, respiratory rate, blood pressure) was performed in all patients for at least 24 h. The continuous electrocardiographic monitoring performed in our stroke unit involves automatic rhythm analysis to detect atrial fibrillation through dedicated software. The diagnosis is subsequently confirmed by a 12-lead electrocardiogram, evaluated by a cardiologist. Furthermore, all patients were also subjected to a 24-h ECG Holter recording, which was evaluated by a cardiologist.

This study was conducted in accordance with the amended Declaration of Helsinki. Local institutional review boards approved the study (protocol number: ID-5137), and written informed consent was obtained from all patients or caregivers.

### 2.2. Statistical Analysis

First, we calculated sample size of the study population. According to the data extrapolated from the literature, we considered an expected prevalence of AF and OSA in the AIS population in an in-hospital setting of 20% [13] and 64% [11], respectively. Therefore, the expected random association of the two conditions is 12%. Based on these assumptions, the minimum number of patients to enroll to have a confidence level of 95% and a margin of error of 5% was 162. To determine if there is an association between AF and OSA in AIS, we compared the expected prevalence of each condition with that observed in the sample. Then, we compared the observed frequency of the association OSA + AF with that expected from a random co-occurrence of the two conditions, by means of Pearson’s χ^2^.

Subsequently, we divided the study population into two subgroups based on the diagnosis of OSA (OSA+ vs. OSA− groups) and compared the two subgroups for clinical and demographic variables. Continuous and categorical variables were summarized using means and standard deviations, and counts and percentages, respectively. Continuous variables were tested using the nonparametric Mann–Whitney U test, while categorical variables were tested using Pearson’s χ^2^. The level of significance was set at *p* < 0.05.

Next, to adjust for potential confounding effects of well-known risk factors for OSA and stroke, variables compared in the univariate analysis were evaluated using backward stepwise binary logistic regression analysis. The dependent variable for the binary logistic regression model was the diagnosis of OSA (OSA+ vs. OSA−), and the predictors were the variables with *p* < 0.25 in the univariate analysis, along with variables considered to be clinically relevant. The goodness of fit for the logistic regression model was evaluated using the Hosmer–Lemeshow test. Odds ratios, *p* values, and 95% confidence intervals are reported.

Finally, we correlated the AHI with the continues variables collected (age, BMI, neck circumference, and NIHSS) and estimated correlation coefficients using Spearman’s rho.

Statistical analysis was performed by means of the Statistical Package for Social Science (SPSS^®^) software, version 20.

## 3. Results

A total of 174 patients were consecutively enrolled (mean age 67.3 ± 11.6 years; 95 males). Demographic and clinical features of the study group are reported in Table 1. OSA and AF were present in 89 (51.2%) and 55 (31.6%) patients, respectively. Therefore, the expected number of patients that should present both conditions, if randomly distributed, was 28 (16.1%). The association of OSA + AF was observed in 33/174 cases (19.0%), which was not statistically different from the expected co-occurrence of the two conditions (χ^2^ = 0.500; *p* = 0.481).

Subsequently, we compared OSA+ (89/174) and OSA− (85/174) groups. Regarding anthropometric measurements, BMI (OSA+: 28.4 ± 6.6 vs. OSA−: 25.9 ± 4.7; *p* = 0.001) and neck circumference (OSA+: 42.3 ± 4.3 vs. OSA−: 39.4 ± 4.4 cm; *p* < 0.001) were significantly higher in the OSA+ population. The prevalence of hypertension (OSA+: 76/89 vs. OSA−: 50/85; *p* < 0.001) and dysphagia (OSA+: 63/89 vs. OSA−: 29/85; *p* < 0.001) was significantly higher in OSA+; the prevalence of diabetes was higher in the OSA+ group, showing a trend toward statistical significance (OSA+: 33/89 vs. OSA−: 20/85; *p* = 0.052). The prevalence of AF was higher, but not statistically different, in the OSA+ group (OSA+: 33/89 vs. OSA−: 22/85; *p* = 0.112). We observed a significantly higher value of C-AHI in the OSA+ group (OSA+: 5.6 ± 3.2 vs. OSA−: 3.2 ± 6.1 events/hour; *p* = 0.028). No other significant differences were observed between the two subgroups; in particular, the prevalence of OSA did not statistically differ according to the stroke subtypes (*p* = 0.483), as shown in Figure 1. Detailed results of the univariate comparison between OSA+ and OSA− patients are reported in Table 2.

In the multivariate binary logistic regression, hypertension (odds ratio = 5.19; 95% confidence interval = 1.58–16.85; *p* = 0.006) and dysphagia (odds ratio = 7.08; 95% confidence interval = 2.23–22.35; *p* < 0.001) were independent predictors of OSA after adjustment for possible confounders. The model of multivariate analysis showed a good fit according to the Hosmer–Lemeshow test (*p* = 0.507). Detailed results of multivariate analysis are reported in Table 3.

Finally, we observed a significant positive correlation between OSA severity, measured by O-AHI, and BMI (Spearman’s rho = 0.289; *p* < 0.001) and neck circumference (Spearman’s rho = 0.329; *p* < 0.001), and stroke severity, measured by NIHSS, and CSA severity, measured by C-AHI (Spearman’s rho = 0.191; *p* < 0.013). No significant correlations were observed between O-AHI and stroke severity and age. Significant correlations are reported in Figure 2.

## 4. Discussion

The current study confirms that OSA and AF are both highly prevalent in AIS. Nevertheless, the frequency of their association is not different from what is expected by the random occurrence of the two conditions. The prevalence of AF in our sample was 31.6%, which is higher than in previous studies investigating the prevalence of AF in AIS. This finding is in line with recent evidence indicating that the prevalence of AF in AIS has been constantly increasing in recent years [14]. These data could be the result of an increased probability of diagnosing AF in the stroke unit due to the prolonged and continuous ECG monitoring. The prevalence of OSA (defined as AHI ≥10 events/h) in our cohort was 51.2%, which is similar to previous studies investigating the prevalence of OSA in AIS [11,15].

The lack of association between OSA and AF in our population suggests that OSA contributes independently to AIS, and that there is no causative relationship between the two conditions in determining stroke. However, it is worth noticing that the OSA+ group presented a non-statistically significant higher prevalence of AF. From this point of view, larger studies with a considerably higher number of patients may demonstrate a causative link between AF and OSA in causing stroke. Notably, we excluded central apneas and hypopneas from the count of O-AHI. We excluded central events because of the well-documented relatively high prevalence of central apneas in both AF and acute stroke [16,17]. Additionally, central apneas recognize a different pathogenic mechanism compared to obstructive apneas [18]. The exclusion of central events in the count of O-AHI could have led to a lower prevalence of OSA in our population regarding patients who presented concomitant heart diseases and AF.

To date, the association of AF and OSA in AIS has been poorly studied, with conflicting findings. Masukhani et al. [19] retrospectively evaluated a population with OSA, reporting a higher prevalence of ischemic stroke in those who presented concomitant AF and proposing an additional role of OSA in the interplay between AF and AIS. However, the study design did not allow for the demonstration of a causative relationship between OSA and AF in determining stroke. Recently, Dalmar et al. [20] observed an increased prevalence of stroke and AF in obese patients with concomitant OSA compared to obese patients without OSA; however, in the stroke group only 20% of patients with stroke and OSA had documented AF, suggesting that the stroke risk is mediated by other factors. Lipford et al. [7] reported an increased prevalence of cardioembolic stroke in an OSA population; however, the relationship between OSA and cardioembolic stroke was still significant after adjusting for known AF, indicating that AF does not entirely justify the association between OSA and stroke. In contrast to what was reported by Lipford et al. [7], we did not observe a significantly higher prevalence of OSA according to the stroke subtypes. However, in our cohort, the prevalence of OSA appears to be particularly high in lacunar strokes, suggesting that OSA plays a determining role in the cerebral small vessel disease [21]. Conversely, Bassetti et al. [22] found an increased prevalence of OSA in strokes due to a large artery disease rather than in cardioembolic stroke, proposing a role of OSA in the atherogenic process. In the Sleep Heart Health Study [23], a strict association between stroke and OSA was still observed after excluding from the analysis patients who presented AF. Munoz et al. [10] found that OSA represented an independent risk factor for AIS in a large elderly population, also after adjusting for others known risk factors, including AF. In addition, OSA has been identified as an independent risk of stroke in patients with AF [24,25], and, particularly, the severity of OSA-related hypoxia has been found to be associated with an increased risk of cardioembolic stroke, stratified by CHA2DS2-VASc [26]. Recent evidence suggests that a thrombogenic atrial substrate can lead to atrial thromboembolism even in the absence of AF [27]. OSA has been linked to several alterations, such as endothelial dysfunction, atrial fibrosis and dilatation, and mechanical dysfunction in the left atrial appendage, all of which have been associated with atrial thrombosis regardless of the presence of AF [27].

Furthermore, it is important to consider that AIS is a condition characterized by autonomic imbalance [28] and an altered respiratory pattern [29]. Therefore, both OSA and AF could be not only determinants but also consequences of AIS [30]. In fact, autonomic imbalance is a frequent complication of AIS, particularly in lesions involving the insula [31]. Animal models have proven a direct role of the autonomic output imbalance in inducing persistent and paroxysmal arrhythmias [32]. Similarly, an altered regulation of the autonomic nervous system has been observed in humans prior to the onset of paroxysmal AF, and interventions aimed at reducing autonomic innervation of the heart result in better control of atrial arrhythmias [33]. Dysautonomia can have a partial or complete recovery after stroke, with autonomic unbalance potentially being transient [34] or persisting for several months [35]. From this perspective, AF could be a transient manifestation of AIS.

Similarly, it is possible that OSA is, at least in part, a reversible manifestation of AIS resulting from an altered respiratory pattern, pharyngeal muscular dysfunction, and prolonged immobilization [18,30,36]. In this view, AIS may exacerbate or precipitate pre-existing sleep-disordered breathing [18]. Current evidence suggests that OSA may be a pre-existing condition aggravated by stroke, while CSA may appear de novo as a symptom of the acute phase. Longitudinal studies evaluating the evolution of sleep apnea after acute stroke have revealed only a slightly lower prevalence of OSA in the chronic phase, and amelioration of CSA [15,22,37,38,39].

Regarding the secondary aim of our study, we observed that patients with AIS and concomitant OSA constitute a subgroup of stroke patients with a higher prevalence of other risk factors for cerebrovascular disease. Specifically, our study showed that OSA patients were more likely to have hypertension and diabetes, indicating a close relationship between these conditions, which promote each other in a detrimental way. As expected, the OSA population presented higher BMI and neck circumference, which are well-established risk factors for OSA. Furthermore, the severity of OSA was directly correlated with higher BMI and neck circumference [40]. Another interesting data point, although not confirmed in the multivariate analysis, was the higher number of central apneas in stroke patients with concomitant OSA. This observation suggests that AIS may result in a unique sleep-related breathing disorder characterized by the simultaneous presence of central and obstructive components [41]. Various mechanisms contribute to sleep-disordered breathing in AIS, including compromised upper airways patency (as a result of the weakness or incoordination of the pharyngeal, intercostal, and diaphragmatic muscles) and reduced arousal response. Additionally, the involvement of the respiratory centers located in the brainstem, caused either by the direct localization of the ischemic lesion or by diffuse cerebral injury resulting from the ischemic stroke, can lead to the development of central apneas.

Finally, we observed a high prevalence of dysphagia in our cohort of stroke patients with OSA. Dysphagia is the commonest clinical manifestation of the involvement of pharyngeal muscles in AIS. Our previous study demonstrated a close association between OSA and dysphagia in AIS, suggesting that pharyngeal muscle palsy represents a common pathogenic link between these two conditions in AIS [42]. In this view, OSA+ patients are at higher risk to develop complications related to dysphagia, in particular aspiration pneumonia. Interestingly, dysphagia is often reversible within a few weeks after stroke, due to gradual recovery of pharyngeal muscle function [43,44]. Therefore, the sleep-disordered breathing observed in AIS may improve simultaneously with the recovery of dysphagia [16,22,45].

Our study has several limitations. First, the observation was limited to patients with acute ischemic stroke, which may not reflect the entire population affected by AF and/or OSA outside of the acute stroke phase. Additionally, the AHI cut-off value for diagnosing OSA in acute stroke is not well established. Our choice to use a cut-off of 10 events/h comes from data available in the literature, although a different cut-off could lead to different prevalence. Furthermore, the diagnosis of AF was based on the analysis of previous clinical recordings or was detected during hospitalization, potentially missing patients with paroxysmal atrial fibrillation. Although we found a nonsignificant higher prevalence of AF in our sample of patients with OSA, larger studies are needed to investigate the potential pathological link between OSA and AF in promoting ischemic stroke. Finally, our study’s cross-sectional design limits the availability of follow-up data, and thus we cannot draw any meaningful conclusions regarding the effect of OSA treatment on AF or stroke outcomes.

## 5. Conclusions

In conclusion, our data suggest that OSA and AF are highly prevalent but not associated in AIS, indicating that they may not share a common pathogenic link in determining stroke. Our findings support that OSA acts as an independent risk factor for stroke, promoting stroke through multifactorial and complex mechanisms beyond a direct association with AF.

## Figures and Tables

**Figure 1 jpm-13-00527-f001:**
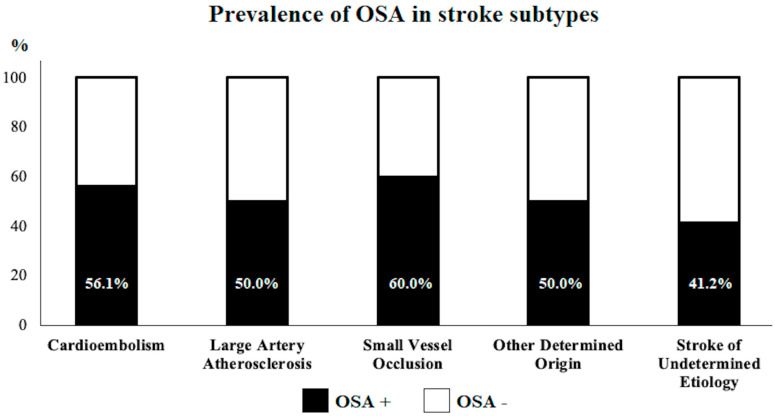
Prevalence of OSA according to the stroke etiology.

**Figure 2 jpm-13-00527-f002:**
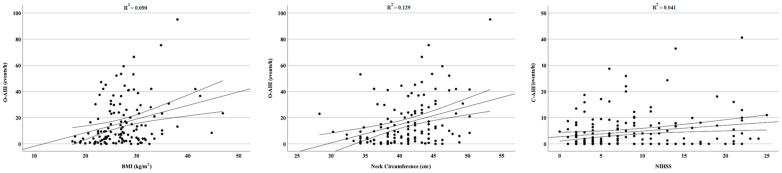
A positive significant correlation was observed between O-AHI and BMI, O-AHI and neck circumference, and C-AHI and stroke severity.

**Table 1 jpm-13-00527-t001:** Demographic features, respiratory indices, and clinical features of the study population.

Demographic Features	
Age, years, mean (SD)	67.3 (11.6)
Gender, male, no. (%)	95 (54.6)
BMI, mean (SD)	27.0 (5.8)
Neck circumference, cm, mean (SD)	40.7 (4.6)
Respiratory Indices	
O-AHI, mean (SD)	17.9 (19.0)
C-AHI, mean (SD)	4.4 (6.7)
ODI, mean (SD)	20.7 (20.5)
Clinical Features	
OSA, no. (%)	89 (51.2)
Atrial fibrillation, no. (%)	55 (31.6)
Hypertension, no. (%)	126 (72.4)
Diabetes, no. (%)	53 (30.5)
Dyslipidemia, no. (%)	76 (43.7)
Reperfusion therapy, no. (%)	38 (21.8)
NIHSS, mean (SD)	8.8 (6.8)
Dysphagia, no. (%)	92 (52.9)
Pneumonia, no. (%)	10 (5.7)
Death, no. (%)	2 (1.1)
Wake-up stroke, no. (%)	50 (28.7)
OCSP	
TACI, no. (%)	42 (24.1)
PACI, no. (%)	81 (46.6)
LACI, no. (%)	20 (11.5)
POCI, no. (%)	31 (17.8)
TOAST	
Cardioembolism, no. (%)	66 (37.9)
Large Artery Atherosclerosis, no. (%)	28 (16.1)
Small Vessel Occlusion, no. (%)	25 (14.4)
Other Determined Origin, no. (%)	4 (2.3)
Stroke of Undetermined Etiology, no. (%)	51 (29.3)

Abbreviations: BMI: body mass index; C-AHI: central apnea–hypopnea index; LACI: lacunar infarcts; NIHSS: National Institutes of Health Stroke Scale; OCSP: Oxfordshire Community Stroke Project classification; ODI: oxygen desaturation index; O-AHI: obstructive apnea–hypopnea index; OSA: obstructive sleep apnea; PACI: partial anterior circulation infarcts; POCI: posterior circulation infarcts; SD: standard deviation; TACI: total anterior circulation infarcts; TOAST: Trial of ORG 10172 in Acute Stroke Treatment (TOAST) classification.

**Table 2 jpm-13-00527-t002:** Univariate comparison between OSA+ and OSA− groups.

	OSA+ (*n* = 89)	OSA− (*n* = 85)	Mann–Whitney U	χ^2^	*p*
Demographic features					
Age, years, mean (SD)	68.6 (11.0)	66.0 (12.2)	4229.5		0.178
Gender, male, no. (%)	49 (55.1)	46 (54.1)		0.015	0.901
BMI, mean (SD)	28.4 (6.6)	25.9 (4.7)	2792.5		**0.001**
Neck circumference, cm, mean (SD)	42.3 (4.3)	39.4 (4.4)	2921.0		**<0.001**
C-AHI, mean (SD)	5.6 (7.2)	3.2 (6.1)	4250.0		**0.028**
O-AHI, mean (SD)	31.8 (17.7)	3.7 (2.9)			
ODI, mean (SD)	34.7 (19.8)	6.6 (7.2)			
Clinical features					
Atrial fibrillation, no. (%)	33 (37.1)	22 (25.9)		2.521	0.112
Hypertension, no. (%)	76 (85.4)	50 (58.8)		17.866	**<0.001**
Diabetes, no. (%)	33 (37.1)	20 (23.5)		3.778	0.052
Dyslipidemia, no. (%)	40 (44.9)	36 (42.4)		0.040	0.842
Reperfusion therapy, no. (%)	22 (24.7)	16 (18.8)		0.262	0.609
NIHSS, mean (SD)	9.3 (6.8)	8.3 (6.9)	4128.5		0.296
Dysphagia, no. (%)	59 (66.3)	33 (38.8)		13.165	**<0.001**
Pneumonia, no. (%)	7 (7.9)	3 (3.5)		0.963	0.327
Death, no. (%)	2 (2.2)	0 (0.0)		1.701	0.192
Wake-up stroke, no. (%)	24 (27.0)	26 (30.6)		0.279	0.359
OCSP				1.944	0.584
TACI, no. (%)	25 (28.1)	17 (20.0)			
PACI, no. (%)	39 (43.8)	41 (48.2)			
LACI, no. (%)	9 (10.1)	12 (14.1)			
POCI, no. (%)	16 (18.0)	15 (16.9)			
TOAST				3.468	0.483
Cardioembolism, no. (%)	37 (56.1)	29 (43.9)			
Large Artery Atherosclerosis, no. (%)	14 (50.0)	14 (50.0)			
Small Vessel Occlusion, no. (%)	15 (60.0)	10 (40.0)			
Other Determined Origin, no. (%)	2 (50.0)	2 (50.0)			
Stroke of Undetermined Etiology, no. (%)	21 (41.2)	30 (58.8)			

Abbreviations: BMI: body mass index; C-AHI: central apnea–hypopnea index; LACI: lacunar infarcts; NIHSS: National Institutes of Health Stroke Scale; OCSP: Oxfordshire Community Stroke Project classification; ODI: oxygen desaturation index; O-AHI: obstructive apnea–hypopnea index; OSA: obstructive sleep apnea; PACI: partial anterior circulation infarcts; POCI: posterior circulation infarcts; SD: standard deviation; TACI: total anterior circulation infarcts; TOAST: Trial of ORG 10172 in Acute Stroke Treatment (TOAST) classification. Bold font indicates statistical significance.

**Table 3 jpm-13-00527-t003:** Multivariate analysis: predictors of OSA.

	Odds Ratio	95% Confidence Interval		*p*
		Lower	Upper	
Age	1.011	0.960	1.064	0.684
Gender (Male)	0.443	0.151	1.298	0.138
BMI	1.182	1.032	1.353	0.016
Neck Circumference	1.116	0.965	1.290	0.138
Wake-up stroke	0.744	0.266	2.083	0.574
NIHSS	0.941	0.856	1.034	0.208
Dysphagia	6.945	1.964	24.552	**0.003**
Hypertension	5.349	1.465	19.534	**0.011**
Diabetes	1.437	0.493	4.186	0.506
Atrial Fibrillation	0.861	0.279	2.657	0.795
C-AHI	1.037	0.959	1.120	0.364

Abbreviations: BMI: body mass index; C-AHI: central apnea–hypopnea index; NIHSS: National Institutes of Health Stroke Scale. Bold font indicates statistical significance.

## Data Availability

Data are available from the corresponding author upon reasonable request.

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
