# Peer review of "Association of Obstructive Sleep Apnea and Atrial Fibrillation in Acute Ischemic Stroke: A Cross-Sectional Study"

_jpm, 2023, doi:10.3390/jpm13030527_

Round 1

Reviewer 1 Report

It would be better to state "prospective cohort study" in the title.

The authors do not have an explanation for the cause of patients with acute ischemic stroke (AIS). It is thought that it is necessary to classify by cause whether embiloc, thrombus, or stenosis is the cause, and to analyze the association according to the classification.

Author Response

We thank the Reviewer for his/her time and useful comments. The text has been edited and corrected.

It would be better to state "prospective cohort study" in the title.

We modified the text following the Reviewer suggestions, specifying that the study has a cross sectional design with a prospective enrollment (Title: Page 1, Line 3; Methods: Page 2, Line 65).

The authors do not have an explanation for the cause of patients with acute ischemic stroke (AIS). It is thought that it is necessary to classify by cause whether embiloc, thrombus, or stenosis is the cause, and to analyze the association according to the classification.

We added stroke etiology based on the TOAST classification to Tables 1 and 2. In comparing OSA+ and OSA- patients, we did not observe significant differences in the prevalence of OSA with respect to stroke etiology (see Tables 1 and 2; Methods: Page 2, Lines 74-76;  Results Section: Page 4, Lines 149-150). Additionally, we included a figure depicting the prevalence of OSA in stroke subtypes (see Figure 1). In our sample, the prevalence of OSA was higher in lacunar strokes even though this difference did not reach statistical significance. This point has been briefly discussed (See Discussion: Page 5, Lines 199-203)

Reviewer 2 Report

Dear

1. There are a lot of grammatical errors.

2. Please add reference (s) for some sentences in methods such as:diagnosis of ischemic stroke with NIHSS ≥ 1 confirmed with neuroimaging, underwent a
polygraphic sleep study within 7 days of AIS onset,
American Academy of Sleep Medicine, if there was a reduction ≥ 30% of the airflow, and so on.

3. Please add abbreviations below the tables.

4. Analysis is very simple. Please add correlations and graphs.

5. Please pay attention to the order of references. See [34, 28, 17].

6. Please delete pronouns in text such as our, my, us, ...

7. Criteria for groups are incomplete.

8. Abstract does not support the text mainly its aims and results.

Author Response

We thank the Reviewer for his/her time and useful comments. The text has been edited and corrected. Further references have been added to the manuscript.

1) There are a lot of grammatical errors.

The text has been extensively corrected..

2) Please add reference (s) for some sentences in methods such as:diagnosis of ischemic stroke with NIHSS ≥ 1 confirmed with neuroimaging, underwent a polygraphic sleep study within 7 days of AIS onset, American Academy of Sleep Medicine, if there was a reduction ≥ 30% of the airflow, and so on.

The following reference has been added: “Berry, R.B.; Budhiraja, R.; Gottlieb, D.J.; Gozal, D.; Iber, C.; Kapur, V.K.; Marcus, C.L.; Mehra, R.; Par-thasarathy, S.; Quan, S.F.; et al. Rules for scoring respiratory events in sleep: update of the 2007 AASM Manual for the Scoring of Sleep and Associated Events. Deliberations of the Sleep Apnea Definitions Task Force of the American Academy of Sleep Medicine. J Clin Sleep Med 2012, 8, 597-619, doi:10.5664/jcsm.2172.” (Methods: Page 2, Line 84).

3) Please add abbreviations below the tables.

We moved the abbreviations below the tables.

4) Analysis is very simple. Please add correlations and graphs.

We thank the Reviewer for the useful suggestion. We implemented our statistical plan as follow:

- multivariate binary regression analysis in order to identify predictors of OSA in patients with acute stroke (see Methods: Page 3, Lines 118-125; Results: Page 4, Lines 153-158; Table 3);

- correlations between AHI and numerical variables (See Methods: Page 3, Lines 126-128; Results: Page 4, Lines 159-164). We added a figure with scatter plots of significant correlations (see figure 2)

- Among variables analyzed, we added stroke etiology defined with TOAST classification (see Tables 1 and 2; Methods: Page 2, Lines 74-76; Results Section: Page 4, Lines 149-150). Additionally, we included a figure depicting the prevalence of OSA in stroke subtypes (see Figure 1). Furthermore, we analyzed central apneas/hypopneas index (C-AHI) (see Tables 1 and 2; Methods: Page 2, Lines 70-71; Results Section: Page 4, Lines 161-162). The higher value of C-AHI in OSA+ group has been discussed (Discussion: Pages 5-6, Lines 244-254).

5) Please pay attention to the order of references. See [34, 28, 17].

We modified the order of references.

6) Please delete pronouns in text such as our, my, us, ...

Done

7) Criteria for groups are incomplete.

We revised the criteria, adding exclusion criteria (see Methods Page 2, Lines 68-69). Moreover, the Methods section has been extensively edited.

8) Abstract does not support the text mainly its aims and results.

The abstract has been edited, including novel findings and mitigating the conclusions.

Reviewer 3 Report

manuscript is nicely drafted, and it address points properly

Author Response

We thank the reviewer for her/his appreciation.

Reviewer 4 Report

The authors examine the association of obstructive sleep apnea and atrial fibrillation after ischemic stroke in a prospective monocentric cohort (n=174). In comparison with the expected prevalence of both sleep apnea and atrial fibrillation, there seemed to be no excess risk of both conditions combined.

There are several restrictions to the study, some of which are mentioned by the authors.

The chosen cut-off for diagnosis of sleep apnea (AHI > 9/h) makes it difficult to compare the rate of sleep apnea in this study with values from the cited meta-analyses.

Atrial fibrillation was diagnozed after 24 h of rhythm monitoring, but without additional Holter monitoring or further strategies to identify atrial fibrillation. It is not clear whether stroke unit monitoring involved software-based, automatic rhythm analysis, as established in many stroke units.

There is no long-term follow-up to corroborate the diagnosis of sleep apnea, as hypopnea / apnea indices tend to peak in the direct aftermath of ischemic stroke and decline thereafter. There is no information on any previous diagnosis of sleep apnea before stroke, either.

The rate of atrial fibrillation is numerically higher in OSA patients (11.2 % difference) which might have reached statistical significance in a larger cohort. This leaves the conclusion that OSA and AF are not interrelated a provisional one. Larger studies would be necessary to disprove the connection between SA and OSA.

There is no information on treatment of OSA and any possible effect on AF incidence.

Author Response

We thank the reviewer for his/her time and useful suggestions. 

The authors examine the association of obstructive sleep apnea and atrial fibrillation after ischemic stroke in a prospective monocentric cohort (n=174). In comparison with the expected prevalence of both sleep apnea and atrial fibrillation, there seemed to be no excess risk of both conditions combined.

There are several restrictions to the study, some of which are mentioned by the authors.

The limitations section has been revised (Page 6, Lines 265-277).

The chosen cut-off for diagnosis of sleep apnea (AHI > 9/h) makes it difficult to compare the rate of sleep apnea in this study with values from the cited meta-analyses.

The selected prevalence (64%) refers to the meta-analysis by Johnson and Johnson cited in the text. In particular, this meta-analysis reveals that the majority of studies assessing the prevalence of OSA in patients with acute-phase stroke in a hospital setting used the AHI cut-off of ≥10 events/hour. For further details see the table 4 of the current paper “Johnson, K.G.; Johnson, D.C. Frequency of sleep apnea in stroke and TIA patients: a meta-analysis. J Clin Sleep Med 2010, 6, 131-137.”

Following the Reviewer observation, we justified the choice of the cut-off of 10 events/h in the text (Page 2, Lines 81-83) and we add in the limitations section the following sentence “the AHI cut-off value for diagnosing OSA in acute stroke is not well-established, and our choice to use a cut-off of 10 events/hour comes from data available in literature, although a different cut-off could lead to different prevalence.” (Page 6, Lines 267-270).

Atrial fibrillation was diagnozed after 24 h of rhythm monitoring, but without additional Holter monitoring or further strategies to identify atrial fibrillation. It is not clear whether stroke unit monitoring involved software-based, automatic rhythm analysis, as established in many stroke units.

We thank the Reviewer to underline this point. In our study all patients were subjected to a 24-hour ECG Holter recording; in fact, in our Stroke Unit 24-hour ECG holter recording is part of the routinely clinical practice. The continuous electrocardiographic monitoring performed in our stroke unit involves the automatic rhythm analysis to detect atrial fibrillation through dedicated software. The diagnosis is subsequently confirmed by a 12-lead electrocardiogram, evaluated by a cardiologist. (Page 2, Lines 92-97).

There is no long-term follow-up to corroborate the diagnosis of sleep apnea, as hypopnea / apnea indices tend to peak in the direct aftermath of ischemic stroke and decline thereafter. There is no information on any previous diagnosis of sleep apnea before stroke, either.

All patients were newly diagnosed with OSA; this point has been elucidated in the inclusion/exclusion criteria in the Methods section (Page 2, Lines 68-69). For what concerns the evolution of SDB, we agree with the Reviewer. Up to date it is not clear in which measure SDB during the acute stroke phase are a cause or a consequence of the stroke. This point has been discussed (Discussion; Page 5, Lines 229-236).

The rate of atrial fibrillation is numerically higher in OSA patients (11.2 % difference) which might have reached statistical significance in a larger cohort. This leaves the conclusion that OSA and AF are not interrelated a provisional one. Larger studies would be necessary to disprove the connection between SA and OSA.

We agree with the reviewer. This point has been elucidated in the discussion (Page 4, Lines 177-180; Page 6, Lines 272-274) and conclusions have been mitigated.

There is no information on treatment of OSA and any possible effect on AF incidence.

Since the study has a cross-sectional design (the study design has been specified in the title and methods section) we cannot draw any meaningful conclusion on this point. This point has been clearly explained in the limitations (see Discussion: Page 6, Lines 275-277).

Round 2

Reviewer 1 Report

It seems to have been revised well.

Author Response

We thank the Reviewer for her/his appreciation